



# Brief communication: Betz's Law: the Zorich Derivation

Richard Zorich

P. O. Box 342 Garberville, CA 95542 USA

**Correspondence:** Richard Zorich
orcid.org/0009-0007-2879-3056
Email: rick4567892000@yahoo.com

**Abstract.** In this article, Betz's law is derived in a new way. A power equation is constructed by accounting for the forces that a machine applies to the air mass that flows through it. By comparing that power equation to the available power in the wind, Betz's law is validated.

**Nomenclature**

$\Delta t$      a time interval

$\Delta v$      a change in velocity

$\rho$      the density of air
Assume that $\rho$ is a constant.

$A$      the area of the machine that is exposed to the wind

$P_m$      the wind power captured by the machine

$P_o$      wind power available to the machine

$T$      the total thrust
$T$ is parallel to $v_o$ and in the same direction as $v_o$.

$T_d$      the part of $T$ due to the wind pulling on the machine from downwind

$T_u$      the part of $T$ due to the wind pushing on the machine from upwind

$v_f$      the final velocity at the end of the wake (see figure number 1)

$v_m$      the velocity of the air flowing through the machine (see figure number 1)
Assume that the flow through each part of $A$ has the velocity $v_m$.
Assume that $v_m$ is parallel to $v_o$ and in the same direction as $v_o$.



$v_o$      the wind velocity

$v_o$ is in a positive direction.

Assume that $v_o$ is in a steady state and has the same velocity everywhere in the atmosphere that is outside the influence of the machine.

# 1 Introduction

Betz's law states that the maximum efficiency of a wind -powered machine is 59%. It was derived over one hundred years ago by the German physicist Albert Betz (1966) and contemporaries. Their derivation has been discussed by van Kuik (2007) and Okulov and van Kuik (2012). They disregard viscosity in order to use Bernoulli's principle in their derivation. Assuming inviscid flow leads to a useful model.

In contrast with Betz's approach, my derivation, recognizes the key role that viscosity plays in the interaction of a wind-
powered machine with the atmosphere. It is my belief that my model gives us better understanding of how a machine captures energy from the wind. I treat a machine as a black box and assume that we have no knowledge of the mechanism. I assume that the machine can capture energy from the wind and that the machine is held in place by a tower so that it doesn't move downwind when it is exposed to the wind.



## 2   MY DERIVATION OF BETZ'S LAW

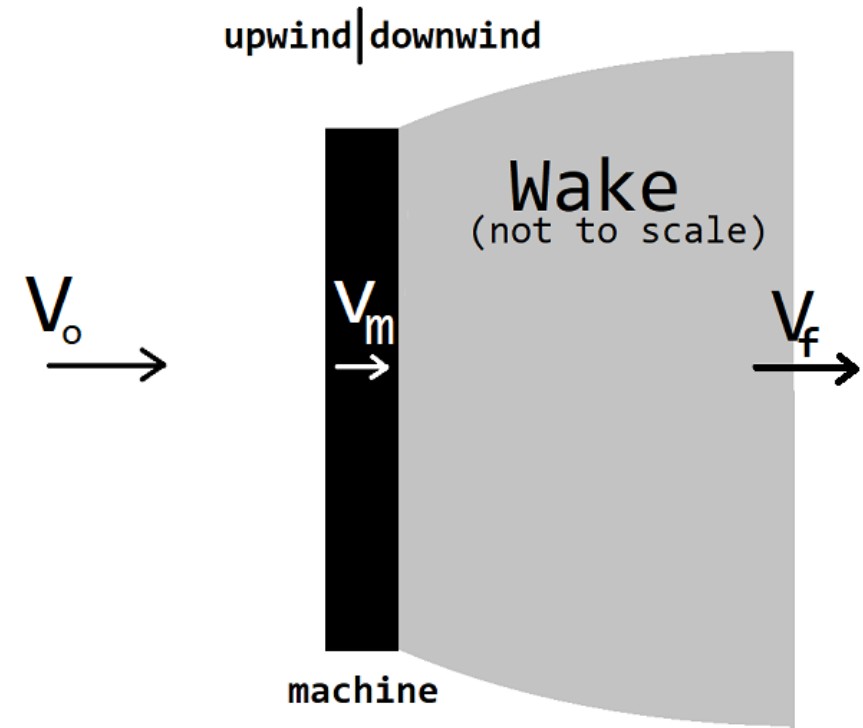

**Figure 1.** machine

$V_o$ is the wind velocity.

$V_m$ is the velocity of the air mass flowing through the machine.

$V_f$ is the velocity of the air mass at the end of the wake.

The wake is not drawn to scale.

Let $v_o$ equal the wind velocity and $v_m$ equal the velocity of the air flowing through the machine (see figure number 1). Imagine that we have complete control of the flow of air through the machine and we can choose to set the value of $v_m$ within the following range.

$$0 \leq v_m \leq v_o \tag{1}$$

If we choose to let

$$v_m = v_o \tag{2}$$

then the machine has no influence and the wind will simply pass through the machine. If we choose to let

$$v_m = 0 \tag{3}$$





then no wind will pass through the machine. If we choose an intermediate value for $v_m$, then only part of the wind directly in front of the machine will flow through it. The other part will be forced to flow around it. Betz's law determines the value of $v_m$

that will maximize the power that a machine captures from the power in the wind.

By convention, the force applied by the wind on the machine that tends to push and pull the machine downwind is called thrust and is denoted by $T$. It is called thrust because historically a wind-powered machine was considered the inverse of a propeller. Thrust is also equal to the force applied by the machine on the air mass that flows through it. This is due to Newton's third law of motion. I divide thrust into two parts. The first part: $T_u$ is applied by the atmosphere that is upwind from the

machine and tends to push it downwind. The second part: $T_d$ is applied by the atmosphere that is downwind from the machine and tends to pull it downwind.

According to Newton's second law of motion

$$\text{Force} = (\text{mass}) \times (\text{acceleration}) \tag{4}$$

Acceleration is defined as the rate of change of the velocity of an object with respect to time.

$$\text{Force} = (\text{mass}) \times \frac{\Delta v}{\Delta t} \tag{5}$$

and with some rearranging, we have

$$\text{Force} = \frac{(\text{mass})}{\Delta t} \times \Delta v \tag{6}$$

where

$$\frac{(\text{mass})}{\Delta t} = (\text{mass flow rate}) \tag{7}$$

thus we have

$$\text{Force} = (\text{mass flow rate})\Delta v \tag{8}$$

The mass flow rate through the machine is equal to the density multiplied by the volumetric flow rate. Thus, we have

$$(\text{mass flow rate}) = \rho A v_m \tag{9}$$

and

$$T = \rho A v_m \Delta v \tag{10}$$

The preceding equation is an expression of Newton's second law. Thrust equals the mass flow rate through the machine times the velocity changes that that mass flow undergoes.

I have divided thrust into two parts,

$$T = T_u + T_d \tag{11}$$





and I will account for the parts separately.

The first part of thrust is a reaction to the force applied to the incoming air mass by the machine. The force that the machine applies, decelerates the air mass that flows through it. The direction of that force is in the negative direction, which is in the opposite direction of thrust. The velocity change experienced by the flow entering the machine from upwind is

$$(v_m - v_o) \tag{12}$$

Therefore, the equation for the upwind part of thrust is

$$-T_u = \rho A v_m (v_m - v_o) \tag{13}$$

and multiplying both sides by a negative one gives

$$T_u = \rho A v_m (v_o - v_m) \tag{14}$$

On the downwind side of the machine, because of the air's viscosity , the wind mixes into the wake and works to accelerate the air mass that flowed through the machine. The second part of thrust, $T_d$, is the force applied by the wind that pulls the air mass through the machine. Let's define (see figure number 1)

$$v_f = \text{(the final velocity at the end of the wake)} \tag{15}$$

The velocity change experienced by the air mass that exited the machine and was pulled downwind through the wake is

$$(v_f - v_m) \tag{16}$$

thus we have

$$T_d = \rho A v_m (v_f - v_m) \tag{17}$$

and since

$$T = T_u + T_d \tag{18}$$

we have

$$T = \rho A v_m [(v_o - v_m) + (v_f - v_m)] \tag{19}$$

The machine is taking energy out of the atmosphere. Therefore, according to the law of conservation of energy , $v_f$ will always be less than $v_o$. But, because the atmosphere is very big and the machine is relatively tiny, $v_f$ will be very close to the value of $v_o$.

Measurements of a wind turbine's wake were taken in the field by Aitken et al. (2014). Their observations indicate that the velocity deficit gradually declines to 15%-25% at a downwind distance of 6.5 rotor diameters. Dong et al. (2022), using large



eddy simulation determined that the velocity in the wind farm wake recovers 95% at 55 rotor diameters downstream. So, it would seem that a wake continues downwind for quite a long distance.

For this discussion, I have assumed that the whole atmosphere has the same velocity except the part that is influenced by the machine. The energy that the machine captures is small compared to the energy contained in the atmosphere. Therefore, I postulate that the length of the wake is finite. We can also imagine that the atmosphere is infinite in size and has an unlimited amount of energy. We can rewrite the last equation by using the concept of a limit . If we assume that

$$v_f \rightarrow v_o \text{ as } A \rightarrow 0 \tag{20}$$

then we have

$$\lim_{A \rightarrow 0} (\rho A v_m (v_o - v_m)) = T_d \tag{21}$$

and

$$\lim_{A \rightarrow 0} (2\rho A v_m (v_o - v_m)) = T \tag{22}$$

The wind pushes and pulls air through the machine. The machine applies a force on the air mass that resists the flow. This is how the machine captures energy from the wind. Power can be defined as

$$\text{Power} = (\text{Force}) \times (\text{Velocity}) \tag{23}$$

Let's define

$$P_m = (\text{the wind power captured by the machine}) \tag{24}$$

thus

$$P_m = T v_m \tag{25}$$

and

$$P_m = \lim_{A \rightarrow 0} (2\rho A v_m^2 (v_o - v_m)) \tag{26}$$

This equation gives us an expression for the machine's power at a certain wind velocity and whatever value of $v_m$ that we allow to flow through the machine. By itself, the amount of power, can't tell us anything about the machine's efficiency. We need to compare the machine's power to the power in the wind.

Wind energy is the kinetic energy of the wind. Kinetic energy is defined as

$$\text{Kinetic Energy} = \frac{1}{2}(\text{mass}) \times (\text{velocity})^2 \tag{27}$$

We can equate power to the flow of kinetic energy. Thus

$$\text{Wind Power} = \frac{(\text{Kinetic Energy})}{\Delta t} \tag{28}$$





$$\text{Wind Power} = \frac{1}{2}\frac{(mass)}{\Delta t} \times (\text{velocity})^2 \tag{29}$$


$$\text{Wind Power} = \frac{1}{2}(\text{mass flow rate}) \times (\text{velocity})^2 \tag{30}$$

If we allow the wind to freely pass through the machine, then the mass flow rate will be equal to $\rho A v_o$ Let's define

$$P_o = (\text{wind power available to the machine}) \tag{31}$$

thus we have

$$P_o = \frac{1}{2}\rho A v_o^3 \tag{32}$$

The mechanical efficiency of the machine is the ratio of the power captured by the machine to the wind power available to it. The power coefficient is a dimensionless number used to express efficiency. It is defined as

$$C_p = \frac{P_m}{P_o} \tag{33}$$

thus, we have

$$C_p = \lim_{A \to 0}\left(\frac{2\rho A v_m^2(v_o - v_m)}{\frac{1}{2}\rho A v_o^3}\right) \tag{34}$$

and

$$C_p = 4\left(\frac{v_m}{v_o}\right)^2\left(1 - \frac{v_m}{v_o}\right) \tag{35}$$

A graph of that equation(see figure number 2) shows that $C_p$ is maximum when

$$\frac{v_m}{v_o} = \frac{2}{3} \tag{36}$$

and the maximum at that value is

$$C_p = 0.5926 \tag{37}$$

0.5926 is known as Betz's coefficient.

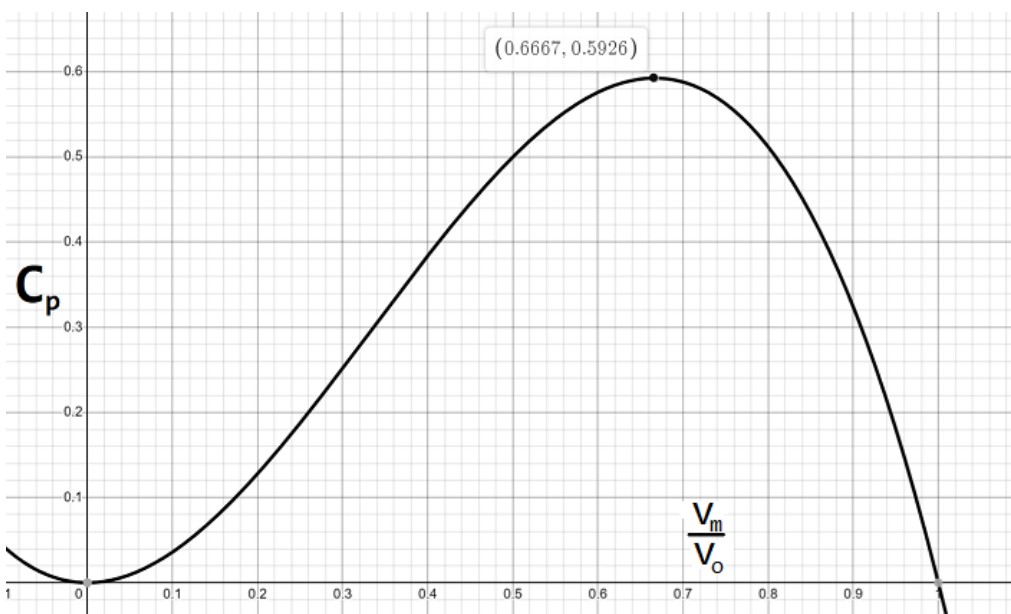

**Figure 2.** Power Coefficient

$C_p$ is the vertical axis.

$V_m/V_o$ is the horizontal axis.

*Author contributions.* For this work, Richard Zorich worked independently on conceptualization, writing original draft preparation, and writing review and editing.

*Competing interests.* The author declares that he has no conflict of interest.



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
