# Peer review of "Brief communication: Betz's Law: the Zorich Derivation"

_Wind Energy Science, 2023_

## Referee Comment (RC1)

The manuscript claims to give a new derivation of Betz' law, English Translation in: WIND ENGINEERING, VOL 37, NO. 4, pp 441-444, (2013). It also claims to give "a better understanding" and "recognizes the key role of viscosity".

A standard text-book-like derivation proceeds like the following: Define a Control Volume which consists of all space filled with air which is influenced by the rotor. Apply integral conservation laws of momentum, energy and mass by respecting the CONTINUUM mechanical nature of the problem which is different from Newton's point mass mechanics. Define power of wind and power extracted of the rotor (machine). As the purpose of such a derivation is to give a save upper limit of possible energy/power extraction, no discussion of influence of viscosity and/or from real atmospheric flow must be included. In addition, a very dangerous statement is about a possible "better understanding". Firstly, understanding (by experience of teaching generations of students) is a very individual psychologic process and secondly "good (better)" and "bad" are expressions from fairy tales and should be used in scientific language.

The author is invited to re-read the papers of van Kuik (DOI: 10.1002/we) and van Kuik, Soerensen and Okulov (DOI: 10.1016/j.paerosci.2014.10.001) and compare it to the many standard text-book derivations. In summary, I do not see and advantage, gain or novelty in this short note and therefore cannot recommend publication.

---

## Referee Comment (RC2)

**Review comments**

It is always interesting if it is possible to come up with alternative formulations of the Betz limit or other ways of deriving it. Unfortunately, the derivation in the present paper is based on a simple sign error and an assumption that is not justified. The problem is that the derivation given in the paper is not consistent. It is too vague just to phrase some general statements regarding the use of Newton's law. In fluid mechanics, as in continuum mechanics in general, it is required to introduce the integral form of the equations with properly defined control volumes and sign conventions. Doing this, the integral equations of mass conservation and axial momentum is conveniently written as (see e.g. Fox and McDonald's text book 'Introduction to fluid mechanics', Wiley and son's):

$$\int_{CV} \rho \vec{U} \cdot d\vec{A} = 0,$$

$$\int_{CV} u_x \rho \vec{U} \cdot \vec{n} dA = \sum F_x,$$

where *CV* denotes the chosen control volume, $u_x$ is the axial velocity, *U=(u_x, u_y, u_z)* is the velocity vector and *dA* denotes the boundaries of the control volume, with $\vec{n}$ designating the outward pointing direction vector.

[Figure]

Example of a control volume

Now, dividing the flow domain into two parts, one going from far upstream of the rotor to a section ending just in front of the rotor and another starting just behind the rotor and ending far downstream, both laterally limited by the stream-surface going through the rotor tips. Using the same notation as in the paper, we get:

Upstream CV analysis: $\quad -\rho A_0 V_0^2 + \rho A_m V_m^2 = \rho A_m V_m (V_m - V_0) = -p_m^+ A_m$

Downstream CV analysis: $-\rho A_m V_m^2 + \rho A_f V_f^2 = \rho A_m V_m (V_f - V_m) = p_m^- A_m$

Here $p_m^+$ is the pressure acting on the upstream surface of the rotor and $p_m^-$ is the pressure acting on the downstream surface of the rotor. From these two equations, the thrust is given as the pressure difference over the rotor multiplied by rotor area:

$$T = (p^+ - p^-)A_m = \rho A_m V_m (V_0 - V_f)$$

The derivation of this equation corresponds to the one in the paper ending with eq. (19). However, in the paper the second downstream equation has a wrong sign, such that this equation incorrectly reads

$$T = (p^+ - p^-)A_m = \rho A_m V_m \left[(V_0 - V_m) - (V_m - V_f)\right].$$

This equation is obviously not correct, and it does not make it more correct by assuming that $V_f = V_0$, which is the second erroneous assumption in the paper. Interestingly, this assumption then results in the same equation as the one derived by Betz, when inserted into eq. (19). However, the derivation is based on a flaw (wrong sign) and an assumption that cannot be justified. Hence, the analysis in incorrect, and I cannot recommend that the paper be published.